# Nutritional Feeding Strategies in Pediatric Intestinal Failure

**DOI:** 10.3390/nu12010177

**Published:** 2020-01-08

**Authors:** Joanne Olieman, Wendy Kastelijn

**Affiliations:** Department of Internal Medicine, Division of Dietetics, Erasmus MC, University Medical Center Rotterdam, Wytemaweg 40, 3015 GD Rotterdam, The Netherlands; w.kastelijn@erasmusmc.nl

**Keywords:** intestinal failure, short bowel syndrome, nutritional feeding strategies, adaptation, blended diet, microbiome

## Abstract

Intestinal failure is defined as a critical reduction of the gut mass or function, below the minimum needed to absorb nutrients and fluids. The ultimate goal in intestinal failure is to promote bowel adaptation and reach enteral autonomy while a healthy growth and development is maintained. The condition is heterogeneous and complex. Therefore, recommendations for the type and duration of parenteral, enteral, and oral nutrition are variable, with the child’s age as an additional key factor. The aim of this review is to provide an overview of nutritional feeding strategies in this heterogeneous population. Different perspectives on nutritional management, nutrition and adaptation, and microbiome and nutrition will be discussed.

## 1. Introduction

Intestinal failure is defined as a critical reduction of the gut mass or function, below the minimum needed to digest and absorb sufficient nutrients and fluids for adequate growth and development in children [1,2]. Intestinal failure (IF) can be caused by several disorders of the gastrointestinal tract, which may be categorized into three main groups; (1) anatomical disorders (e.g., short bowel syndrome) (2) neuromuscular diseases (e.g., chronic intestinal pseudo-obstruction) and (3) mucosal intestinal diseases (e.g., microvillous inclusion disease) [1]. Short bowel syndrome (SBS) is the main cause of IF, accounting for at least 40% of the cases [3,4,5]. According to the Dutch National Working group on SBS, short bowel syndrome is defined as ≥70% resection of the small bowel and/or the residual small bowel length distal to ligament of Treitz of <50 cm in premature neonates, <75 cm of term neonates, and <100 cm in children ≥1 year [6]. SBS may result from an extensive resection due to congenital defects such as intestinal atresia and gastroschisis, or postnatal ischemia of the bowel from necrotizing enterocolitis or volvulus [5,7,8]. The reduction of the absorptive and digestive surface and the resulting decreased availability of digestive enzymes and transport proteins causes malabsorption in SBS patients [9]. Furthermore, some underlying disorders such as gastroschisis and intestinal atresia not only affect the remaining bowel length, but may also have impact on the residual bowel’s (motor) function and adaptation potential [10]. In clinical practice SBS can be discerned into three phases. The first phase occurs directly after resection and is characterized by diarrhea and massive loss of fluids and electrolytes, causing increased risk of dehydration, and impaired gut motility also occurs [11]. Shortly after resection the remaining part of the bowel attempts to increase its fluid and nutrient absorption [12]. This is the second phase where the process of bowel adaptation begins and can last up to one to four years [1,13]. It includes muscular hypertrophy (increased bowel diameter and wall thickness) and mucosal hyperplasia [1]. The adaptation process is more pronounced in the ileum than in the jejunum. Bowel adaptation processes should lead to full intestinal autonomy. The third phase is the so-called resting state in which a status quo has arisen regarding the tolerance of enteral nutrition and hence the need for parenteral nutrition. It is characterized by permanent degree of malabsorption, there might be an electrolyte imbalance, bone demineralization, and kidney or gallstones might occur. Vitamin and mineral deficiencies are frequently seen in this phase. 

The clinical manifestation of SBS is determined by the residual functional length of the jejunum and ileum, the presence of an enterostomy, the presence (or absence) of the ileocecal valve, the remaining functional length of the colon, underlying pathology, and possible complications [14]. These factors affect bowel adaptation and therefore the functionality of the gastrointestinal tract, which in turn affects feeding options. Therefore, recommendations for the type and duration of parenteral, enteral, and oral nutrition are variable, with the child’s age as an additional key factor. 

The goals of nutritional support in patients with SBS are twofold—providing safe, adequate supplemental nutrients to preserve lean body mass and function, and if possible, supporting and accelerating the body’s own adaptive mechanisms [15]. Maintaining simultaneously optimal nutritional status and achieving intestinal adaptation is a clinical challenge in patients with SBS. Both growth and development of the child as well as bowel adaptation should be considered synergistically as primary outcome parameters [11].

The aim of this review is to provide an overview of nutritional feeding strategies in this heterogeneous population.

## 2. Nutritional Management According to Phases SBS

In order to meet the nutritional needs of the child, it is initially necessary to provide parenteral nutrition (PN). Clinical experience shows that PN is sometimes given in addition to enteral nutrition for a long period of time in order to be able to meet the energy and nutrient requirements. Sometimes PN is provided for several years and some children remain on PN for life.

During the acute phase PN is indispensable and often fluid and electrolyte losses need to be additionally compensated. During this phase minimal enteral feeding should be started as soon as possible to promote bowel adaptation [11].

The aim of PN therapy is to provide the nutritional requirements for normal growth and development, while the bowel undergoes the adaptation necessary for the transition to an enterally based diet. During the adaptation phase enteral nutrition should be introduced, preferably orally, in order to stimulate oral motor activity and to avoid feeding aversion behavior. When advancements in enteral nutrition are successful, PN should be decreased and cycled. The aim of cycling PN is to reduce hyperinsulinemia, with subsequent fat accumulation and liver disease [16,17,18,19]. 

During the resting state enteral/oral feeding should be maximized and the patient should be weaned of PN while preserving optimal growth. Supplementing deficient micronutrients are often necessary [11]. 

## 3. Adaptation

Intestinal adaptation is a natural compensatory process where structural and functional changes in the intestine occur after bowel resection to improve nutrient and fluid absorption in the remnant bowel [13]. The degree of adaptation potential is dependent on the residual bowel length and the type and quality of the residual bowel. For example, the adaptation capacity of the ileum is greater than in the jejunum [20]. Even though the surface area of the jejunum is greater and this is where the absorption of most nutrients takes place, the jejunum’s ability to reabsorb water and electrolytes is poor. Furthermore, chyme moves slower in the ileum, due to the motility patterns and proximity of the ileocecal valve [21]. The ileum absorbs bile salts, vitamin B12, fat-soluble vitamins, and electrolytes. When the ileum is lacking, the risk of vitamin deficiencies is high.

Adaptation includes the structural changes such as muscular hypertrophy (increased bowel diameter and wall thickness), mucosal hyperplasia (increased cell proliferation), and angiogenesis (formation new blood vessels) [13]. These changes support mucosal growth and thereby potentially absorption function. The functional changes are increased expression of transporter proteins and nutrient/electrolyte exchangers, as well as accelerated crypt cell differentiation and deceased transit time which results in increased nutrient and fluid absorption [13]. However, most of these changes are seen in animal models and it is not yet fully understood how all of these mechanisms contribute to intestinal adaptation in humans [13].

Enteral nutrition plays an important role in the adaptation process: the luminal nutrients are known to have a stimulating effect on the epithelial cells and the production of trophic hormones. Thereby intraluminal nutrients also increase pancreatic and biliary secretion [7,12,22]. Enteral nutrients not only prevents mucosal atrophy, but also loss of barrier function and downregulation of the mucosal immune system [23]. The mechanism by which enteral nutrients stimulate adaptation is complex and can be broken down into three major categories: (1) by stimulation of mucosal hyperplasia through direct contact with epithelial cells; (2) stimulation of trophic gastrointestinal hormone secretion; and (3) stimulation of the production of trophic pancreatobiliary secretions [12,24].

It has been postulated that the higher the complexity of a nutrient, the higher the workload of the digestive mechanism involved. Thus, the more digestion a nutrient (e.g., whole protein) needs, the more hyperplasia it will cause [25]. In other words, this “functional workload,” which challenges the digestive and absorptive function of the remnant bowel, is key to its adaptation [21]. The composition of the diet should be considered in an effort to balance gastrointestinal tolerance with specific nutrients in a complex form that may further stimulate the adaptive process [26]. Digestion and absorption of macronutrients have not been well characterized and the most appropriate dietary constituents vary per patient. 

### 3.1. Protein

Dietary proteins are either directly digested into amino acids and absorbed, or digested into polypeptides, which are first absorbed inside the enterocytes before they are hydrolyzed to amino acids [27]. Dietary protein hydrolysates have been developed to optimize both absorption pathways [11]. There are three types of formulas: whole protein-based, hydrolyzed protein-based, and amino acid-based formula. There is no evidence that hydrolyzed protein formulas are better tolerated than whole protein-based formulas [28]. Whole proteins are preferred in terms of increased workload to the digestive and absorptive function of the bowel. Hydrolyzed proteins appear to empty the stomach faster than whole proteins and consequently elicited a more rapid increase in plasma amino acid, glucagon, and insulin concentrations in enterally fed surgical adult patients [29,30]. A few small case studies described how the amino acid-based formula reduced PN requirement and reduced allergies [31,32]. These studies were small and had no control group. One amino acid that received a lot of attention is glutamine. Glutamine is the main fuel for enterocytes [33] and is thought to enhance mucosal hyperplasia [34,35]. However, glutamine supplementation in infants with gastrointestinal disease showed no difference in PN dependence, feeding tolerance, and intestinal absorptive-barrier function compared to those who received a placebo amino acid [36].

### 3.2. Carbohydrates

Disaccharides are more trophic than monosaccharides due to the higher functional workload of the bowel [21]. Lactose intolerance may occur in patients with proximal jejunum resection [25]. A cross-over study in adults with SBS, however, demonstrated a similar tolerance of a lactose-free diet compared to a diet containing 20 g of lactose a day [37]. Since there is not enough evidence, a lactose-free diet is not recommended in patients with SBS [25]. Moreover, lactose may promote the production of short chain fatty acids (SFCAs) in the colon and a small amount of lactose is therefore recommended for infants [11]. Dietary fiber can be divided into soluble and insoluble forms. Insoluble forms, such as cellulose (e.g., in cereals) bind to water and cause bulking and softening of the stool. Soluble forms such as pectin and guar gum (found in fruit and vegetables) slow gastric emptying and the overall gut transit time, resulting in a mild anti-diarrheal effect [22,38]. Soluble fiber and some starches pass undigested into the colon, where they are fermented by colonic bacteria into SFCA’s, which account for 5–10% of the energy intake [39]. Atia et al. showed that starch is the primary carbohydrate substrate for colonic bacterial fermentation in patients with SBS, although pectin also enhances SFCA production and fluid absorption [40]. Butyrate (SCFA) has trophic effects on the jejunal and ileal cells when delivered in the colon [41]. The suggested underlying mechanism of this trophic effect may be the stimulated release of glucagon-like peptide 2 (GLP-2) [42]. Some animal studies showed that pectin enhanced bowel adaptation [43,44]. Only one case study described that pectin supplementation in a single patient prolonged transit time and enhanced nitrogen absorption [45]. 

### 3.3. Fat

Long chain triglycerides (LCTs) undergo bile-dependent hydrolysis within the enterocyte before export into the lymphatic system as chylomicrons [25]. It has been suggested that LCTs enhance bowel adaptation [46]. A few animal studies showed that lipids were effective in enhancing bowel adaptation [47,48]. The presence of LCT stimulates the secretion of PYY and GLP-2 which mediates the ileal and jejunal brake, resulting in slower transit time [49]. LCTs contain n-3 long chain polyunsaturated fatty acids (n-3 LCPs), which might have beneficial effects in patients with SBS [11]. Besides their properties for brain development, they are also known to have anti-inflammatory effects and improve the splanchnic circulation [50,51]. More studies are needed to confirm these beneficial effects [11]. A few small studies showed that enteral n-3 LCPs improved cholestasis in infants with SBS [52,53]. Medium chain triglycerides (MCTs) are directly absorbed across the enterocyte into the portal circulation [25]. This process starts in the stomach. Theoretically this absorption process would increase fat uptake and provide a faster energy supply [11]. An RCT in patients with jejuno- or ileostomy showed that a diet containing high concentrations of MCTs can cause osmotic diarrhea as a result of rapid hydrolysis of MCTs [54]. In contrast, a study in patients with an intact colon showed that MCTs improved fat absorption and therefore might be beneficial in patients with bile acid or pancreatic insufficiency [55]. However, it should be kept in mind that MCTs are saturated fats and do not contain essential fatty acids.

### 3.4. Non-Nutrient Luminal Factors

Luminal factors include a variety of nutrients, secretions, and other essential components in the diet or produced in the lumen of the gastrointestinal tract that have been known to stimulate gut mucosal growth. GLP-2 is produced in the enteroendocrine cells of the terminal ileum and colon in response to luminal nutrients (especially LCTs and carbohydrates). It has been shown that GLP-2 increases the mucosal surface area of the gut, slows motility, up-regulates nutrient absorption, improves gut-barrier function, and increases intestinal blood flow [56]. Serum levels of GLP-2 increase when adaptation progresses [57]. GLP-2 analogues are nowadays used as a treatment option for improving nutrients and fluid. A few studies in children with IF have shown that GLP-2 therapy resulted in a consistent increase in nutrient absorption, reduction in PN requirement, and adequate growth [58,59]. This therapy could affect the feeding strategy. More studies are needed to confirm these positive effects and to establish how permanent intestinal autonomy may be achieved while continuing GLP-2 analog administration. 

### 3.5. Human Milk

The health benefits of human milk have been amply documented; its use is associated with significantly decreased risks of infection, allergies, respiratory diseases, diabetes, and otitis media [60]. Exclusively breastfed children have reduced risk of infectious diseases such as diarrhea and respiratory infections [61]. It has been postulated that human milk, which contains glutamine and growth factors (e.g., growth hormone and epidermal growth factor), might also enhance bowel adaptation [7,12]. A few cohort studies have demonstrated that human milk contains high amounts of nucleotides, immunoglobulin A, and leucocytes, which support the immune system of the neonate [62,63]. Therefore it might be hypothesized that the immunoglobulins and antimicrobial peptides of human milk also promotes intestinal colonization with appropriate lactobacilli and related bacteria, which are important elements of the healthy microbiome [64]. The composition of human milk is complex and changes dynamically over the lactation period [65]. The composition is influenced by gestational age at birth and postnatal age and it can actively accelerate the development of the infant’s own defenses [66]. The protein content of human milk decreases during the first month of lactation. As a consequence, the amino acid content of human milk also varies during the early phase of lactation [65]. The main source of carbohydrates in human milk is lactose and complex oligosaccharides [65]. Some animal studies suggested that bovine colostrum is beneficial to bowel adaptation [67]. However, human studies could not confirm this beneficial effect [68,69]. One study found that breastfed infants with SBS were weaned off PN earlier than non-breastfed SBS infants [70]. Another study in patients with gastroschisis showed that exclusive human milk feeding decreased the time to full enteral autonomy and length of hospital stay [71]. In infants with intestinal failure, human milk is considered to be the first choice [11,72].

### 3.6. Feeding Mode

Nutrient and fluid absorption are dependent on mucosal surface area as well as contact time. Therefore, gastrointestinal motility plays an important role in nutrient absorption [40]. It has been hypothesized that slow continuous infusion of enteral nutrition increases mucosal contact time and thereby enhances absorption by maximizing saturation of carrier proteins [25]. A study in children showed that when enteral nutrition was administered continuously, the intestinal absorption and weight gain improved [73]. A case study in adult SBS patients showed that when continuous enteral nutrition was started early, enteral autonomy could be attained a mean of 36 days after surgery [74]. A small randomized cross-over trial in 15 adult SBS patients showed the continuous enteral nutrition improved energy and macronutrient absorption compared to oral diet [75]. However, bolus feeding causes cyclical changes in plasma levels of gastrointestinal hormones, such as insulin, pancreatic polypeptides, enteroglucagon, motilin, and neurotensin, which might be important for adaptation, growth, and motility [76,77]. Bolus feeding and the ensuing period of fasting maintain normal motility patterns. The fasting motility period is hallmarked by phase III of migrating motor complex and reflects neuromuscular function [11]. Oral feeding activates salivary glands which stimulates the secretion of epidermal growth factor (EGF) and other trophic factors in the saliva, which might enhance adaptation. A few animal studies showed that salivary EGF and vascular endothelial growth factor are important contributing factors in adaptation process [78,79]. Oral feeding should be introduced as soon as possible to stimulate the oral motor development and to avoid feeding aversion behavior [72].

## 4. Nutritional Management 

The goal of nutritional management is to optimize intestinal adaptation and reach intestinal autonomy while preserving adequate growth and development and avoiding complications. The composition, volume, and timing of enteral feeding can affect the achievement of enteral autonomy [77]. Nutritional management differs per goal and should be tailored to the individual patient. Macro- and micronutrient requirements and tolerance vary with age, nutritional status, the remaining bowel length and function, and presence of ileocecal valve and colon [21]. The optimal enteral feeding regimen in children with SBS is still debated by clinicians. Subjects of debate are mode of administration (continuous versus bolus feedings), time of introduction in general, composition (polymeric, semi-elemental, or elemental), time of introduction and composition of oral feeding, and the supplementation of fibers. Most data on enteral nutrition in children with SBS are derived from outcomes of retrospective observational studies and/or case reports [7]; relevant high quality randomized controlled (clinical) trials are scarce. A structured literature review [72] showed that only two evidence-based recommendations could be made; (1) enteral nutrition should be initiated as soon as possible (i.e., a few days after bowel resection) to promote intestinal adaptation [80,81], (2) human milk or standard polymeric formula (depending on age) is the recommended initial feed [28,70]. Other clinical experience recommendations were; (a) enteral nutrition should be administered in a continuous fashion, (b) bottle-feeding (small volumes) should be started as soon as possible in neonates to stimulate suck and swallow reflexes [1,82], (c) solid foods may be introduced at the age of 4–6 months (if necessary, corrected for gestational age) to stimulate oral motor activity and to avoid feeding aversion behavior [1,7,82,83]. Table 1 provides an age-appropriate general overview of an oral/enteral feeding strategy in SBS patients. Oral and/or enteral nutrition should be started in a small volume and increased slowly based on tolerance. Especially when a child is older, there are several ways to introduce and increase enteral/oral feeding. Allowing the child to eat and drink themselves is preferred over solely liquid enteral nutrition. It is advised to make one change at a time and offer small amounts of food frequently. Even when a small volume of enteral feeding is tolerated, cycling PN should be considered in order to minimize the risk of complications [84,85]. The common strategy is to trial 1 h off and 23 h on infusion. The PN rate should be tapered in the last hour of administration to 50% for 30 min, then 25% of the original rate for the final 30 min to limit the risk of hypoglycemia. The PN can be reduced by a further hour every few days aiming for 12–16 h infusion when the first hour is tolerated [86,87]. 

Capriati et al. published a recent update on the current knowledge on the impact of initial diet in neonatal-onset SBS on the process of bowel adaptation [88]. They found 10 eligible studies (mostly retrospective) with a total of 822 patients who were categorized in three groups: hydrolyzed formula-fed patients, patients with a combination of hydrolyzed formula and human milk, and patients with a combination of amino acid-based formula and human milk. Unfortunately, they were not able to form a group with exclusively human milk-fed SBS patients. They found that the course of bowel adaptation, in terms of enteral adaptation rate and PN duration, was not significantly impacted by the initial diet [88]. They concluded that human milk remains the first step in feeding SBS patients and that hydrolyzed formula was not more effective in promoting intestinal adaptation than amino acid-based formula, therefore both types of formulas could be used as a second step [88]. Unfortunately, high-quality research on feeding strategies in SBS patients remain scarce; this is partially due to the small number and the heterogeneity of these patients.

## Blended Diet

Blended diet refers to the use of blended family foods administered into an enteral feeding tube. Currently, interest is growing in the use of blended diet for management of feeding difficulties, reflux, and improved bowel function in the pediatric population [89,90]. It is perceived as more natural, “healthier,” and better tolerated compared to commercially available enteral formulas [90,91]. A few studies in children dependent on enteral nutrition reported that blended diet improved clinical outcomes (e.g., gagging, vomiting, tolerance) compared to commercially available enteral formulas [89,92]. Samela et al. transitioned 10 pediatric intestinal failure patients >1 year of age from an elemental formula to a formula with real food ingredients. Ninety percent tolerated the transition to a 100% blended diet in a mean of 60 days [93]. Around 80% of these patients had their colon in continuity. Blended diet caused more formed and less frequent stools and had appropriate weight gain after one year on a blended diet [93]. The BLEND study showed that 17 out of 20 chronically ill tube feeding-dependent children could successfully be transitioned to a blended diet in four weeks. Even though they needed 50% more calories than commercially available formulas, the prevalence of vomiting and use of acid-suppressive agents significantly decreased [92]. Stool consistency and frequency remained unchanged, but the bacterial diversity and species richness significantly increased [92]. A blended diet has a variable composition and it is difficult to reach sufficient calories in a certain volume. Furthermore, a blended diet is labor intensive and there are concerns regarding hygiene, short shelf life, and increased risk of tube clogging.

More studies are needed to provide evidence that a blended diet is a safe and effective feeding strategy in children with intestinal failure.

## 5. Microbiome

Clinical manifestations of feeding intolerance, such as abdominal distension, bloating, and nausea caused by bacterial overgrowth, are some of the complications that can occur in SBS patients [94]. Lactobacilli tolerate acidic surroundings, and reduced absorption of nutrients provides more substrate for lactobacilli. They produce lactic acid from mono- and disaccharides and may proliferate in the acidic environment resulting in bacterial overgrowth. The human gastrointestinal tract contains around 10^14^ microorganisms (including bacteria) which are commonly referred to as the gut microbiome [95]. The majority of the microorganisms reside within the distal parts of the gastrointestinal tract [96]. The development of the gut microbiome occurs primarily during infancy [97]. Colonization of the gastrointestinal tract starts directly after birth and its composition is dependent on the gestational age, mode of delivery, and type of feeding [98,99]. Breastfed infants tend to have more Actinobacteria and Firmicutes and less Bacteroides than formula-fed infants [100]. During the first years of life, the gut is colonized gradually and the large compositional variation is determined by genetics, environmental factors, diet, and the development of the immune system [101]. During adulthood the composition of the microbiome stabilizes. The microbiome is important for several functions such as fermentation (producing SCFA), nutrient absorption in the colon, development of the immune system, and intestinal mucosal growth and integrity of the gut [102]. It has been described that Clostridia is important for normal intestinal function and protection of intestinal diseases, whereas Enterobacteriaceae are proinflammatory and might be harmful [103,104,105]. Where symbiosis of the gut microbiome causes immune tolerance, intestinal homeostasis, and healthy metabolism, dysbiosis of the microbiome might cause immune and/or allergic diseases, as well as intestinal and metabolic diseases [97].

### 5.1. Microbiome Intestinal Failure and Influencing Factors

A recent review by Neelis et al. showed a nice overview on the evidence on gut microbiome, its metabolic activity, and its association with disease characteristics in adult and pediatric IF patients. They found 10 human studies (50% pediatric) and the most consistent finding of these studies was the overall reduction in bacterial diversity [101]. Intestinal failure patients had a distinctive increase of Proteobacteria (among others (a.o.) Enterobacteriaceae) to 40% compared to 9% in healthy controls. Furthermore, the Bacteroidetes decreased to 19%, whereas in healthy controls these were 46%. An overabundance of Lactobacilli were found in IF patients [101]. Engstrand, Lilja et al. found that the microbiome diversity was significantly reduced in SBS children receiving PN compared to those who were weaned off PN [106], while in the latter group of patients the diversity was still lower than in healthy controls [106]. Neelis et al. concluded that the gut microbiome could potentially be used as biomarker to guide clinical practice during intestinal adaptation, as well as a modifiable therapeutic agent [101]. 

### 5.2. Diet and Microbiome

Factors that influence the microbiome are gastrointestinal factors, such as underlying disease, anatomy (remaining length and type of bowel, motor function, presence of jejuno/ileostomy, absence ileocecal valve), and feeding tubes. Next to that, medication such as antibiotics and proton pump inhibitors play a role in altering the microbiome. Lastly, also nutritional factors, such as composition, fiber intake, type of feeding, administration route, and mode of feeding and the amount, influence the microbiome [101]. The microbiome contributes to host health through biosynthesis of vitamins and essential amino acids, as well as generation of metabolic byproducts from undigested dietary components [96]. One of the most important bacterial metabolites is SCFA, which is an end-product of fermentation of non-digestible dietary carbohydrates (e.g., fiber) by anaerobe bacteria. Even though fiber is the main fuel for production of SCFA, dietary protein, glycoprotein, and peptides, as well as intestinal cell turnover, may also be substrate for fermentation [107]. Ninety-five percent of the SCFAs are absorbed in the colon and may account for 5–10% energy intake [39]. Moreover, SCFAs also stimulate vascular flow and motility and increase sodium absorption [108]. It has been shown that diet can alter the human microbiome even within 24 h after a change in diet and reversion was also shown 48 h after discontinuation of the diet [109]. A review of Singh et al. showed that consumption of plant-based protein caused an increase in Bifidobacterium and Lactobacillus and a decrease in Bacteroides and Clostridium. Increased levels of SCFAs were also observed [96]. Consumption of animal proteins caused an increase in Bacteroides and Biophilia and decrease in Bifidobacterium and, ultimately, a decrease in SCFAs [96]. Both types of protein caused an increased abundance of microbial diversity. Fibers/prebiotics and resistant starch have a positive effect on bacterial abundance, causing an increase of a.o. Lactobacilli and Bifidobacteria [96]. 

These results suggest that a great variation in composition and a balanced diet contribute to a healthy microbiome. 

Most commercially available formulas that are prescribed in SBS patients are animal protein based, which might not contribute to a diverse microbiome. Furthermore, hydrolyzed formula (HF) and amino acid-based formula (AA) often do not contain any fibers. Moreover, clinicians tend to stick to one type of formula and consequently no variation in enteral feeding/diet is provided. Gallagher et al. showed that when patients were transitioned from a commercial formula to blended diet (varied composition and containing more protein [plant and animal] and fiber) the bacterial diversity and species richness significantly increased. The Proteobacteria were significantly reduced and a trend was seen of an increase in Firmicutes [92]. Human milk is dynamic and its composition changes over time, possibly contributing to a diverse microbiome.

Preliminary results of the PREDICT I study showed that identical twins who share genes and most of their environment had different glucose, insulin, and triglycerides responses on identical foods [110]. Furthermore, the twins only shared 37% of their gut microbes; this is only slightly higher than the 35% shared between unrelated individuals. The proportions of nutrients (protein, carbohydrates, and fat) listed on food labels explained less than 40% of the differences between individuals’ nutritional responses to meals with similar calories. In addition, large differences in responses to the same meals were seen depending on the time of day they were eaten [110]. So personal differences in metabolism due to factors such as gut microbiome, meal timing, and exercise are just as important as the nutritional composition of foods.

## 6. Concluding Remarks

Pediatric intestinal failure is a challenging condition to treat. An understanding of intestinal physiology, adaptation process, altering microbiome, and the effect of nutrition on these processes allows for the development of targeted, evidence-based feeding strategies. Due to the multifaceted condition, multidisciplinary care (surgeons, gastroenterologists, dieticians, specialized nurses) for these patients is required. Long-term monitoring of intestinal failure patients even after PN weaning is necessary due to the greater risk of nutrient deficiencies [111,112]. At this moment it is not possible to present evidence-based nutritional management because studies of efficacy of different feeding strategies are scarce and the methodological quality of published data is low. However, a careful nutritional strategy that improves intestinal adaptation in SBS patients who are PN dependent is crucial to reduce the duration of PN and consequently the risk of occurrence of PN-related complications [88]. In light of the adaptation process, early start of minimal feeding is essential and human milk is the preferred choice in neonates. Oral nutrition is not only important for oral motor skills, but may also be beneficial for adaptation. In addition, enteral nutrition (either continuous or bolus) may help to stimulate adaptation and weaning from PN. However, aggressive enteral feeding should be avoided. Age appropriate solid foods may improve consistency in the stool. Furthermore, a variated and balanced diet composition is important for the gut microbiome. In that light, human milk and a blended diet might be a good strategy. Nutritional management should be tailor-made, and it is all about balance and timing—choosing the right food at the right time.

## Figures and Tables

**Table 1 nutrients-12-00177-t001:** Age appropriate oral/enteral feeding strategy.

	Neonate 100% Milk	Infant 4–12 Months Weaning from Milk	Toddler Solid Foods	School Child Solid Foods
Type nutrition	Human milk/polymeric HF/AA	Human milk Puree Solids (e.g., bread) EN: Polymeric/HF/AA	Solid foods EN: Polymeric/HF/AA	Solid foods EN: Polymeric/HF/AA
Mode	Oral and partial enteral	Oral and partial enteral	Oral and partial enteral	Oral and optionally partial enteral
Administration mode	Day Portions Night continuous	Day Portions Night continuous	Day Portions Night continuous	Day Portions Night continuous

HF; hydrolyzed formula, AA; amino acid-based formula, EN; enteral nutrition.

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
