# Peer review of "Nutritional Feeding Strategies in Pediatric Intestinal Failure"

_nutrients, 2020, doi:10.3390/nu12010177_

Round 1

Reviewer 1 Report

Thank you for your comprehensive review on nutritional feeding strategies in the pediatric intestinal failure. 

This seems to be a good handbook for clinicians who should care the patients with intestinal failure, esp. short bowel syndrome. Here I want to recommend a few suggestions to the authors.

Firstly, I want to recommend to re-generate the table 1. There was little information about the feeding strategy according to the age group. There are much more to be added in the table. For example, guidance for the deciding volume of oral or enteral feeding according to the patient's age in a day could be added to the table. Cyclic PN strategy could be pointed out also when partial enteral feeding was considered.  

Secondly, for the future perspective, brief summary for the glucagon-like peptide II  (GLP-2; tedglutide) could be added for the readers. It is known that enteral autonomy could be achieved more easier, if GLP-2 was available for the patients with short bowel syndrome. So, with GLP-2, feeding strategy could be affected. Clinical strategy for the intestinal failure should deal with GLP-2, even though this article focusses just on the nutritional feeding strategy alone.

Thirdly, review for the gut microbiota seems to be too long. Most of them were about basic gut microbiota researches, not directly about intestinal failure. 

Thank you again for your efforts. 

Author Response

Dear reviewer,

Thank you for your valuable comments, we believe that it has improved our manuscript.

Below you will find our response point by point:

Firstly, I want to recommend to re-generate the table 1. There was little information about the feeding strategy according to the age group. There are much more to be added in the table. For example, guidance for the deciding volume of oral or enteral feeding according to the patient's age in a day could be added to the table. Cyclic PN strategy could be pointed out also when partial enteral feeding was considered.The purpose of the table was to provide a general overview of enteral/oral feeding strategy according to age. Adding more information in that table would make it confusing to our opinion. However due the valid points you have raisen,  we have added more clarifying sentences in the manuscript (lines-245-255).   Secondly, for the future perspective, brief summary for the glucagon-like peptide II  (GLP-2; tedglutide) could be added for the readers. It is known that enteral autonomy could be achieved more easier, if GLP-2 was available for the patients with short bowel syndrome. So, with GLP-2, feeding strategy could be affected. Clinical strategy for the intestinal failure should deal with GLP-2, even though this article focusses just on the nutritional feeding strategy alone. thank you for this valid point, we have added a non-nutrient luminal factor  section to the manuscript (line 169-181)  Thirdly, review for the gut microbiota seems to be too long. Most of them were about basic gut microbiota researches, not directly about intestinal failure.  We have reduced the section.

Reviewer 2 Report

Excellent review!

this is a review article, so many of the questions don't seem relevant. I have addressed them below.  

What is the main question addressed by the research? What is the best feeding regimen for pediatric patients with intestinal failure.

Is it relevant and interesting?  Yes

How original is the topic? Not very

What does it add to the subject area compared with other published material? It is a summary of already published material Is the paper well written? Yes
Is the text clear and easy to read? Yes

Are the conclusions consistent with the evidence and arguments presented? Yes

Do they address the main question posed? Yes

Author Response

thank you for the nice comments.

We have checked our spelling and language.
